# Patched Diffusion Models for Unsupervised Anomaly Detection in Brain MRI

**Finn Behrendt**[1]                                                                FINN.BEHRENDT@TUHH.DE
**Debayan Bhattacharya**[1]                                            DEBAYAN.BHATTACHARYA@TUHH.DE
**Julia Krüger**[2]                                                      JULIA.KRUEGER@JUNG-DIAGNOSTICS.DE
**Roland Opfer**[2]                                                      ROLAND.OPFER@JUNG-DIAGNOSTICS.DE
**Alexander Schlaefer**[1]                                                          SCHLAEFER@TUHH.DE

[1] *Institute of Medical Technology and Intelligent Systems, Hamburg University of Technology, Hamburg, Germany*

[2] *Jung Diagnostics GmbH, Hamburg, Germany*

**Editors:** Accepted for publication at MIDL 2023

## Abstract

The use of supervised deep learning techniques to detect pathologies in brain MRI scans can be challenging due to the diversity of brain anatomy and the need for large, pixel-level annotated data sets. An alternative approach is to use unsupervised anomaly detection, which only requires sample-level labels of healthy brain anatomy to create a reference representation. This reference representation can then be compared to unhealthy brain anatomy in a pixel-wise manner to identify abnormalities. To accomplish this, generative models are needed to create anatomically consistent MRI scans of healthy brains. While recent diffusion models have shown promise in this task, accurately generating the complex structure of the human brain remains a challenge. In this paper, we propose a method that reformulates the generation task of diffusion models as a patch-based estimation of healthy brain anatomy, using spatial context to guide and improve reconstruction. We evaluate our approach on data of tumors and multiple sclerosis lesions and demonstrate a relative improvement of 25.1% in segmentation performance compared to existing baselines.

## 1. Introduction

Over the last decades, significant effort has been put into developing support tools that can assist radiologists in assessing medical images (Kawamoto et al., 2005). Convolutional neural networks (CNNs) have proven successful in this task due to their ability to process images effectively (Shen et al., 2017). However, supervised approaches that use CNNs have limitations, such as the need for large amounts of expert-annotated training data and the challenge of learning from noisy or imbalanced data (Ellis et al., 2022; Karimi et al., 2020; Johnson and Khoshgoftaar, 2019).

Unsupervised anomaly detection (UAD) is an alternative approach that can be trained with healthy samples only, eliminating the need for pixel-level annotations. During training, UAD models typically focus on reconstructing images from a healthy training distribution. When unseen, unhealthy anatomy is encountered at test time, high values in the pixel-wise reconstruction error indicate abnormalities.

Recently, denoising diffusion probabilistic models (DDPM) (Ho et al., 2020) have emerged as a state-of-the-art approach for image generation. As a result, they have also been applied

to the problem of unsupervised anomaly detection (UAD) in brain MRI (Wyatt et al., 2022; Pinaya et al., 2022a). DDPMs work by adding noise to an input image, then using a trained model to remove the noise and estimate or reconstruct the original image. Hence, in contrast to most autoencoder-based approaches, DDPMs preserve spatial information in their hidden representation of the input which is important for the image generation process (Rombach et al., 2022). However, applying noise to the entire image at once can make it difficult to accurately reconstruct the complex structure of the brain. To address this issue, we introduce patched DDPMs (pDDPMs) for UAD in brain MRI. In pDDPMs, we apply the forward diffusion process only on a small part of the input image and use the whole, partly noised image in the backward process to recover the noised patch. At test time, we use the trained pDDPM to sequentially noise and denoise a sliding patch within the input image and then stitch the individual denoised patches to reconstruct the entire image.

We evaluate our method on the public BraTS21 and MSLUB data sets and show that it significantly ($p < 0.05$) improves the tumor segmentation performance.

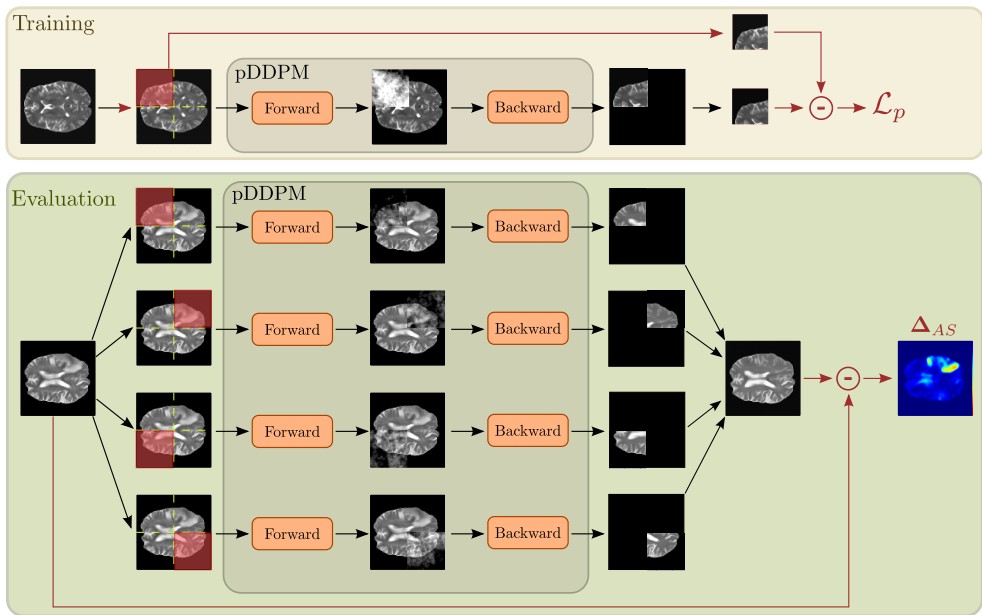

Figure 1: Schematic drawing of our method. From left to right: A patch is sampled within the input image, noise is added to that patch in the forward process and removed in the backward process. During evaluation, we stitch all patches and calculate the pixel-wise error as anomaly map $\mathbf{\Delta}_{AS}$.

## 2. Recent Work

In recent research on UAD in brain MRI, various architectures have been examined. Autoencoders (AE) and variational autoencoders (VAE) have demonstrated reliable training and fast inference, but their blurry reconstructions have hindered their effectiveness in UAD, as noted in (Baur et al., 2021). Therefore, research often focuses on understanding

the image context better by adding spatial latent dimensions (Baur et al., 2018), multi-resolution (Baur et al., 2020b), skip connections together with dropout (Baur et al., 2020a), or a denoising task as regularization (Kascenas et al., 2022). Similarly, modifications to VAEs aim to enforce the use of spatial context by spatial erasing (Zimmerer et al., 2019) or utilizing 3D information (Bengs et al., 2021; Behrendt et al., 2022). Other approaches propose restoration methods (Chen et al., 2020), uncertainty estimation (Sato et al., 2019), adversarial autoencoders (Chen and Konukoglu, 2018) or the use of encoder activation maps (Silva-Rodríguez et al., 2022). Also, vector-quantized VAEs have been proposed (Pinaya et al., 2022b). As an alternative to AE-based architectures, generative adversarial networks (GANs) have been applied to the problem of UAD (Schlegl et al., 2019). However, the unstable training nature of GANs makes their application very challenging. Furthermore, GANs suffer from mode collapse and often fail to preserve anatomical coherence (Baur et al., 2021). To alleviate this, inpainting approaches have been proposed that use the generator to inpaint erased patches during training (Nguyen et al., 2021). Lately, DDPMs have shown to be a promising approach for the task of UAD in brain MRI as they have scalable and stable training properties while generating sharp images of high quality (Wolleb et al., 2022; Wyatt et al., 2022; Sanchez et al., 2022; Pinaya et al., 2022a). While these approaches aim to estimate the entire brain anatomy at once, patch-based DDPMs have been proposed for image restoration (Özdenizci and Legenstein, 2023) and image inpainting (Lugmayr et al., 2022) in the domain of generic images. Patch-based DDPMs are a promising approach also for brain MRI reconstruction, as global context information about individual brain structure and appearance could be incorporated while estimating individual patches. However, current patch-based approaches either neglect the surrounding context of each patch (Özdenizci and Legenstein, 2023) or reconstruct patches from a fully noised image, which also impacts the surrounding context (Lugmayr et al., 2022). Thus, it is of interest to develop patch-based DDPMs that consider both the individual patch and its unperturbed surrounding context for the task of UAD in brain MRI.

## 3. Method

We apply the diffusion process of DDPMs in a patch-wise fashion, meaning that given the input image $x \in \mathbb{R}^{C,W,H}$ with $C$ channels, width $W$ and height $H$, we add noise to a patch $p_k \in \mathbb{R}^{C,h,w}$ with $h < H, w < W$ and $k = [1, ..., K]$. Subsequently, we reconstruct the patch to achieve a local estimate of the brain anatomy. Hereby, our motivation is a better understanding of image context by denoising image patches based on their unperturbed surrounding. Furthermore, we hypothesize that this would also lead to better anatomical coherence in the overall reconstruction of individual brains. As at test time anomalies can appear anywhere in the brain, we need to add and remove noise to the whole brain anatomy with our patch-wise approach. Therefore, we use a sliding window approach where we subsequently add noise to and remove noise from individual patches at positions that are evenly spaced across the image. Having covered the entire input image, we stitch all individual patch reconstructions into one image. This strategy allows estimating each local region in the input by using the spatial context of its surrounding which is assumed to be particularly helpful if the patch covers an anomaly. Our approach is shown in Figure 1.

### 3.1. DDPMs

In DDPMs, first, the image structure is gradually destroyed by noise and subsequently, the reverse denoising process is learned. During the forward process, adding noise $\epsilon \sim \mathcal{N}(\mathbf{0}, \mathbf{I})$ to $\boldsymbol{x}_0$ follows a predefined schedule $\beta_1, ..., \beta_T$:

$$\boldsymbol{x}_t \sim q(\boldsymbol{x}_t|\boldsymbol{x}_0) = \mathcal{N}(\sqrt{\bar{\alpha}_t}\boldsymbol{x}_0, (1-\bar{\alpha}_t)\mathbf{I}), \text{with } \bar{\alpha}_t = \prod_{s=0}^{t}(1-\beta_t). \tag{1}$$

The time step $t$ is sampled from $t \sim Uniform(1, ..., T)$ and controls how much noise is added to $\boldsymbol{x}_0$. For $t = T$ the image is replaced by pure Gaussian noise $\boldsymbol{x}_t = \epsilon \sim \mathcal{N}(\mathbf{0}, \mathbf{I})$ and for $t = 0$, $\boldsymbol{x}_t$ becomes $\boldsymbol{x}_0$.

In the backward process, the goal is to reverse the forward process and to recover $\boldsymbol{x}_0$.

$$\boldsymbol{x}_0 \sim p_\theta(\boldsymbol{x}_t) \prod_{t=1}^{T} p_\theta(\boldsymbol{x}_{t-1}|\boldsymbol{x}_t), \text{with } p_\theta(\boldsymbol{x}_{t-1}|\boldsymbol{x}_t) = \mathcal{N}(\boldsymbol{\mu}_\theta(\boldsymbol{x}_t, t), \boldsymbol{\Sigma}_\theta(\boldsymbol{x}_t, t)). \tag{2}$$

$\boldsymbol{\mu}_\theta$ and $\boldsymbol{\Sigma}_\theta$ are estimated by a neural network with parameters $\theta$. Following (Ho et al., 2020), we use an Unet (Ronneberger et al., 2015) for this task and keep $\boldsymbol{\Sigma}_\theta(\boldsymbol{x}_t, t) = \frac{1-\alpha_{t-1}}{1-\alpha_t}\beta_t\mathbf{I}$ fixed. To derive a tractable loss function, the variational lower bound (VLB) is used. By applying reformulations, Bayes rule and by conditioning $q(\boldsymbol{x}_{t-1}|\boldsymbol{x}_t)$ on $\boldsymbol{x}_0$, minimizing the VLB can be approximated by the simpler loss derivation $\mathcal{L}_{simple} = ||\epsilon - \epsilon_\theta(\boldsymbol{x}_t, t)||^2$. In this work, we utilize the $l1$-error and change the objective to directly estimate $\boldsymbol{x}_0^{rec} \sim p_\theta(\boldsymbol{x}_0|\boldsymbol{x}_t, t)$, leading to $\mathcal{L}_{rec} = |\boldsymbol{x}_0 - \boldsymbol{x}_0^{rec}|$. For sampling images with DDPMs, typically step-wise denoising is applied for all time steps starting from $t = T$. As this comes at the cost of long sampling times, in this work we directly estimate $\boldsymbol{x}_0^{rec} \sim p_\theta(\boldsymbol{x}_0|\boldsymbol{x}_t, t)$ at a fixed time step $t_{test}$. This simplification is possible since we do not aim to generate new images from noise but are interested in reconstructing a given image.

### 3.2. Patched DDPMs

As aforementioned, with patched DDPMs, we apply the forward and backward process in a patched fashion. During training, we sample the patches either at random positions or from a fixed grid defined as follows. We partition $\boldsymbol{x}$ into $K$ patch regions that are evenly spaced across $\boldsymbol{x}$. The number of possible patch regions in $\boldsymbol{x}$ is derived as $K = \lceil\frac{W-w}{w}\rceil + \lceil\frac{H-h}{h}\rceil + 2$, where $\lceil.\rceil$ denotes the ceiling operation. From this grid, we uniformly sample an index $k$.

During the forward step of the diffusion process, i.e., the noising step we sample the noised image $\boldsymbol{x}_t$ only at the given patch position $\boldsymbol{p_k}$. Consider $\boldsymbol{M_p} \in \mathbb{R}^{C,H,W}$ a binary mask where the pixels that overlap with $\boldsymbol{p_k}$ are set to one and pixels that do not overlap with $\boldsymbol{p_k}$ are set to zero. We obtain the partly noised image as

$$\tilde{\boldsymbol{x}_t} = \boldsymbol{x}_t \odot \boldsymbol{M_p} + \boldsymbol{x}_0 \odot \neg\boldsymbol{M_p} \tag{3}$$

where $\odot$ denotes element-wise multiplication. In the backward process, $\tilde{\boldsymbol{x}_t}$ is fed to the denoising network to estimate the given noise area. The denoised image is derived as $\tilde{\boldsymbol{x}}_0^{rec} \sim p_\theta(\boldsymbol{x}_0|\tilde{\boldsymbol{x}_t}, t)$. To train the patch-wise denoising task, we optionally use an objective function $\mathcal{L}_p$ adapted from $\mathcal{L}_{rec}$, where we calculate $\mathcal{L}_p = |(\boldsymbol{x}_0 - \tilde{\boldsymbol{x}}_0^{rec}) \odot \boldsymbol{M_p}|$ based on the noised region within $\boldsymbol{p_k}$, ignoring the surrounding area.

During Evaluation, for every $k \in [0, ..., K]$, we subsequently perform the diffusion process

based on the patch $\boldsymbol{p_k}$. After the reconstruction of all patch regions, we use the reconstructed patches $[\boldsymbol{p}_0^{rec}, ..., \boldsymbol{p}_K^{rec}]$ and stitch them with respect to their original position in the input image to retain the full reconstruction of $\boldsymbol{x}_0$. In the case of overlapping patches, we average the overlapping regions of the reconstructed patches.

## 4. Experimental setup

### 4.1. Data

We use the publicly available IXI data set as healthy reference distribution for training. The IXI data set consists of 560 pairs of T1 and T2-weighted brain MRI scans, acquired in three different hospital sites. From the training data, we use 158 samples for testing and partition the remaining data set into 5 folds of 358 training samples and 44 validation samples for cross-validation and stratify the sampling by the age of the patients.

For evaluation, we utilize two publicly available data sets, namely the Multimodal Brain Tumor Segmentation Challenge 2021 (BraTS21) data set (Baid et al., 2021; Bakas et al., 2017; Menze et al., 2014), and the multiple sclerosis data set from the University Hospital of Ljubljana (MSLUB) (Lesjak et al., 2018).

The BraTS21 data set consists of 1251 brain MRI scans of four different weightings (T1, T1-CE, T2, FLAIR). We split the data set into an unhealthy validation set of 100 samples and an unhealthy test set of 1151 samples. The MSLUB data set consists of brain MRI scans of 30 patients with multiple sclerosis (MS). For each patient T1, T2, and FLAIR-weighted scans are available. We split the data into an unhealthy validation set of 10 samples and an unhealthy test set of 20 samples. For both evaluation data sets, expert annotations in form of pixel-wise segmentation maps are available.

Across our experiments, we utilize T2-weighted images from all data sets. To align all MRI scans we register the brain scans to the SRI24-Atlas (Rohlfing et al., 2010) by affine transformations. Next, we apply skull stripping with HD-BET (Isensee et al., 2019). Note that these steps are already applied to the BraTS21 data set by default. Subsequently, we remove black borders, leading to a fixed resolution of $[192 \times 192 \times 160]$ voxels. Lastly, we perform a bias field correction. To save computational resources, we reduce the volume resolution by a factor of two resulting in $[96 \times 96 \times 80]$ voxels and remove 15 top and bottom slices parallel to the transverse plane.

### 4.2. Implementation Details

We evaluate our proposed method *pDDPM*, against multiple established baselines for UAD in brain MRI. These include *AE*, *VAE* (Baur et al., 2021), their sequential extension *SVAE* (Behrendt et al., 2022), and denoising AEs *DAE* (Kascenas et al., 2022). We also compare simple thresholding *Thresh* (Meissen et al., 2022), and the GAN-based *f-AnoGAN* (Schlegl et al., 2019). Additionally, we chose *DDPM* (Wyatt et al., 2022) as a counterpart to our proposed method. We implement all baselines based on their original publications with the following individual adaptations that have been shown to improve training stability and performance. For *VAE* and *SVAE*, we set the value of $\beta_{VAE}$ to 0.001. For *f-AnoGAN*, we set the latent size to 128 and the learning rate to $1e - 4$.

For *DDPM* and *pDDPM*, we utilize structured simplex noise, rather than Gaussian noise,

as it is known to better capture the natural frequency distribution of MRI images (Wyatt et al., 2022). For training, we uniformly sample $t \in [1, T]$ with $T = 1000$, and at test time, we choose a fixed value of $t_{test} = \frac{T}{2} = 500$. We choose a linear schedule for $\beta_t$, ranging from $1e - 4$ to $2e - 2$ and use an Unet similar to (Dhariwal and Nichol, 2021) as a denoising network. For each channel dimension $C_f \in [128, 128, 256]$, the Unet consists of a stack of 3 residual layers and downsampling convolutions. This structure is mirrored in the upsampling path with transposed convolutions. Skip connections connect the layers at each resolution. In each residual block, groupnorm is used for normalization and SiLU (Elfwing et al., 2018) acts as activation function before convolution. For time step conditioning, the time step is first encoded using a sinusoidal position embedding and then projected to a vector that matches the channel dimension. This is added to the feature representation using scale-shift-norm (Perez et al., 2018) in each residual block. Unless specified otherwise, all models are trained for a maximum of 1600 epochs, and the best model checkpoint, as determined by performance on the healthy validation set, is used for testing. We process the volumes in a slice-wise fashion, uniformly sampling slices with replacement during training and iterating over all slices to reconstruct the full volume at test time. The models were trained on NVIDIA V100 GPUs (32GB) using Adam as the optimizer, a learning rate of $1e - 5$, and a batch size of 32. The code for this work is available at https://github.com/FinnBehrendt/patched-Diffusion-Models-UAD.

## 4.3. Post-Processing and Anomaly Scoring

During training, all models aim to minimize the $l1$ error between the input and its reconstruction. At test time, we use the reconstruction error as a pixel-wise anomaly score $\mathbf{\Delta}_{AS} = |\boldsymbol{x}_0 - \boldsymbol{x}_0^{rec}|$, where high values indicate larger reconstruction errors and vice versa. Given the hypothesis that the models will fail to reconstruct unhealthy brain anatomy, we assume that anomalies are located at regions of high reconstruction errors. We apply several post-processing steps that are commonly used in the literature (Baur et al., 2021; Zimmerer et al., 2019). Before binarizing $\mathbf{\Delta}_{AS}$, we use a median filter with kernel size $K_M = 5$ to smooth $\mathbf{\Delta}_{AS}$ and perform brain mask eroding for 3 iterations. Having binarized $\mathbf{\Delta}_{AS}$, we apply a connected component analysis, removing segments with less than 7 voxels. To achieve a threshold for binarizing $\mathbf{\Delta}_{AS}$, we perform a greedy search based on the unhealthy validation set where the threshold is determined by iteratively calculating Dice scores for different thresholds. The best threshold is then used to calculate the average Dice score on the unhealthy test set (DICE). Furthermore, we report the average Area Under Precision-Recall Curve (AUPRC) and report the mean absolute reconstruction error ($l1$) of the test split from our healthy IXI data set.

## 4.4. Statistical Testing

For significance tests, we employ a permutation test from the MLXtend library (Raschka, 2018) with a significance level of $\alpha = 5\%$ and 10,000 rounds of permutations. The test calculates the two models' mean difference of the Dice scores for each permutation. The resulting p-value is determined by counting the number of times the mean differences were equal to or greater than the sample differences, divided by the total number of permutations.

Table 1: Comparison of the evaluated models with the best results highlighted in bold. *fixed sampling* denotes that patch positions are sampled from a fixed grid, in contrast to *random sampling*, where patch positions are randomly sampled. $\mathcal{L}_p$ denotes calculating the reconstruction loss only on the patch region whereas $\mathcal{L}_{rec}$ denotes calculating the reconstruction loss for the whole image. For all metrics, mean $\pm$ standard deviation across the different folds are reported.

| | BraTS21 | | MSLUB | | IXI |
|---|---|---|---|---|---|
| **Model** | **DICE [%]** | **AUPRC [%]** | **DICE [%]** | **AUPRC [%]** | **$l1$ ($1e-3$)** |
| *Thresh* (Meissen et al., 2022) | 19.69 | 20.27 | 6.21 | 4.23 | 145.12 |
| *AE* (Baur et al., 2021) | 32.87±1.25 | 31.07±1.75 | 7.10±0.68 | 5.58±0.26 | 30.55±0.27 |
| *VAE* (Baur et al., 2021) | 31.11±1.50 | 28.80±1.92 | 6.89±0.09 | 5.00±0.40 | 31.28±0.71 |
| *SVAE* (Behrendt et al., 2022) | 33.32±0.14 | 33.14±0.20 | 5.76±0.44 | 5.04±0.13 | 28.08±0.02 |
| *DAE* (Kascenas et al., 2022) | 37.05±1.42 | 44.99±1.72 | 3.56±0.91 | 5.35±0.45 | **10.12±0.26** |
| *f-AnoGAN* (Schlegl et al., 2019) | 24.16±2.94 | 22.05±3.05 | 4.18±1.18 | 4.01±0.90 | 45.30±2.98 |
| *DDPM* (Wyatt et al., 2022) | 40.67±1.21 | 49.78±1.02 | 6.42±1.60 | 7.44±0.52 | 13.46±0.65 |
| *pDDPM + random sampling + $\mathcal{L}_{rec}$* | 44.47±2.34 | 48.84±2.71 | 9.41±0.96 | 9.13±1.13 | 14.08±0.77 |
| *pDDPM + fixed sampling + $\mathcal{L}_{rec}$* | 47.81±1.15 | 52.38±1.17 | **10.47±1.27** | **10.58±0.85** | 12.12±0.76 |
| *pDDPM + fixed sampling + $\mathcal{L}_p$* | **49.00±0.84** | **54.07±1.06** | 10.35±0.69 | 9.79±0.4 | 11.05±0.15 |

## 5. Results

Unless stated otherwise, for *pDDPM*, we use patch dimensions of $h = w = \frac{H}{2} = \frac{W}{2} = 48$. The comparison of our pDDPM with the baseline models is shown in Table 1. Like *DAE*, the *DDPM* shows relatively high performance on the BraTS21 data set, but its performance on the MSLUB data set is moderate. In contrast, our *pDDPM* outperforms all baselines on both data sets regarding DICE and AUPRC, with statistical significance for the BraTS21 data set ($p < 0.05$). Considering the reconstruction quality by means of $l1$ error on healthy data, the *DAE* shows the lowest reconstruction error, followed by *pDDPM*.

Qualitatively, we observe smaller reconstruction errors from *pDDPMs* compared to *DDPMs* for healthy brain anatomy as shown in Figure 2. Examples of reconstructions from other baseline models can be found in Appendix 4. As seen in Figure 3, a patch size of $60 \times 60$ pixels results in the best performance. Additionally, there is a peak in performance when the noise level at test time is $t_{test} = 400$. A visualization of different noise levels is provided in Appendix B and ablation studies for the MSLUB data set are available in Appendix C.

| Input | $\mathbf{\Delta}_{AS}$ DDPM | $\mathbf{\Delta}_{AS}$ pDDPM | Ground Truth |
|---|---|---|---|

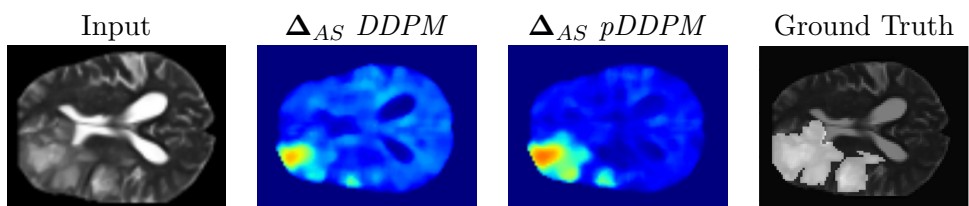

Figure 2: Visualization of input, errormap and the ground truth for *DDPM* and *pDDPM* for the Brats21 data set.

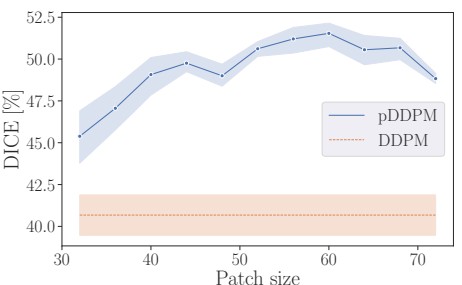 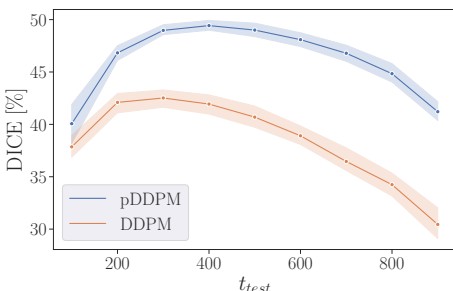

Figure 3: DICE for different patch sizes (left) and noise levels at test time $t_{test}$ (right) for the BraTS21 data set. We report the average DICE across the 5 cross-validation folds. Standard deviations are visualized as enveloping intervals.

## 6. Discussion & Conclusion

Our approach frames the reconstruction of healthy brain anatomy as patch-based denoising, allowing to incorporate context information about individual brain structure and appearance when estimating brain anatomy. We show that *pDDPMs* outperform both their non-patched counterparts and various baseline methods with significant differences for the BraTS21 data set ($p < 0.05$).

Our results indicate that the image context around the noised patch can be used effectively by the model to replace potential anomalies covered by noise patches with estimates of healthy anatomy. From the performance improvements resulting from selecting patches from fixed positions and minimizing $\mathcal{L}_p$ rather than $\mathcal{L}_{rec}$, we conclude that it is helpful to focus on pre-defined local patches during training. By stitching the individual patches, we achieve sharp reconstructions without the downside of reconstructing too much unhealthy anatomy. Note that this trade-off is influenced by both, the noise level $t_{test}$ and the patch size as shown in Figure 3. While our initial values for these hyper-parameters already show robust performance improvements across both data sets, further tuning results in more optimal settings for certain anomalies. To enhance generalization across different anomalies, employing an ensemble of different patch sizes and noise levels, as demonstrated in (Graham et al., 2022), is a promising direction for future research. Evaluating the reconstruction quality by means of $l1$ error, *DAE* shows superior results to *pDDPM*. However, *DAE* is able to reconstruct unhealthy anatomy which increases false negative predictions and thus decreases the UAD performance. We observe that accurately identifying MS lesions in T2-weighted MRI scans is challenging, and the limited number of samples makes it hard to achieve statistically significant results. However, our *pDDPMs* show promising improvements on the MSLUB data set, suggesting that it could be useful to address the challenges of detecting MS lesions. To further improve the UAD performance, using FLAIR-weighted MRI scans or enriching the anomaly scoring by structural differences could be valuable.

Our proposed approach has shown promising results in terms of UAD performance, however, it does have the drawback of an increase in inference time. While parallel computing could alleviate the increase in inference time, future work could focus on guiding the denoising process by spatial context more efficiently.

## Acknowledgments

This work was partially funded by grant number KK5208101KS0 and ZF4026303TS9 and by the Free and Hanseatic City of Hamburg (Interdisciplinary Graduate School) from University Medical Center Hamburg-Eppendorf

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

## Appendix A. Exemplary reconstructions for all Baselines

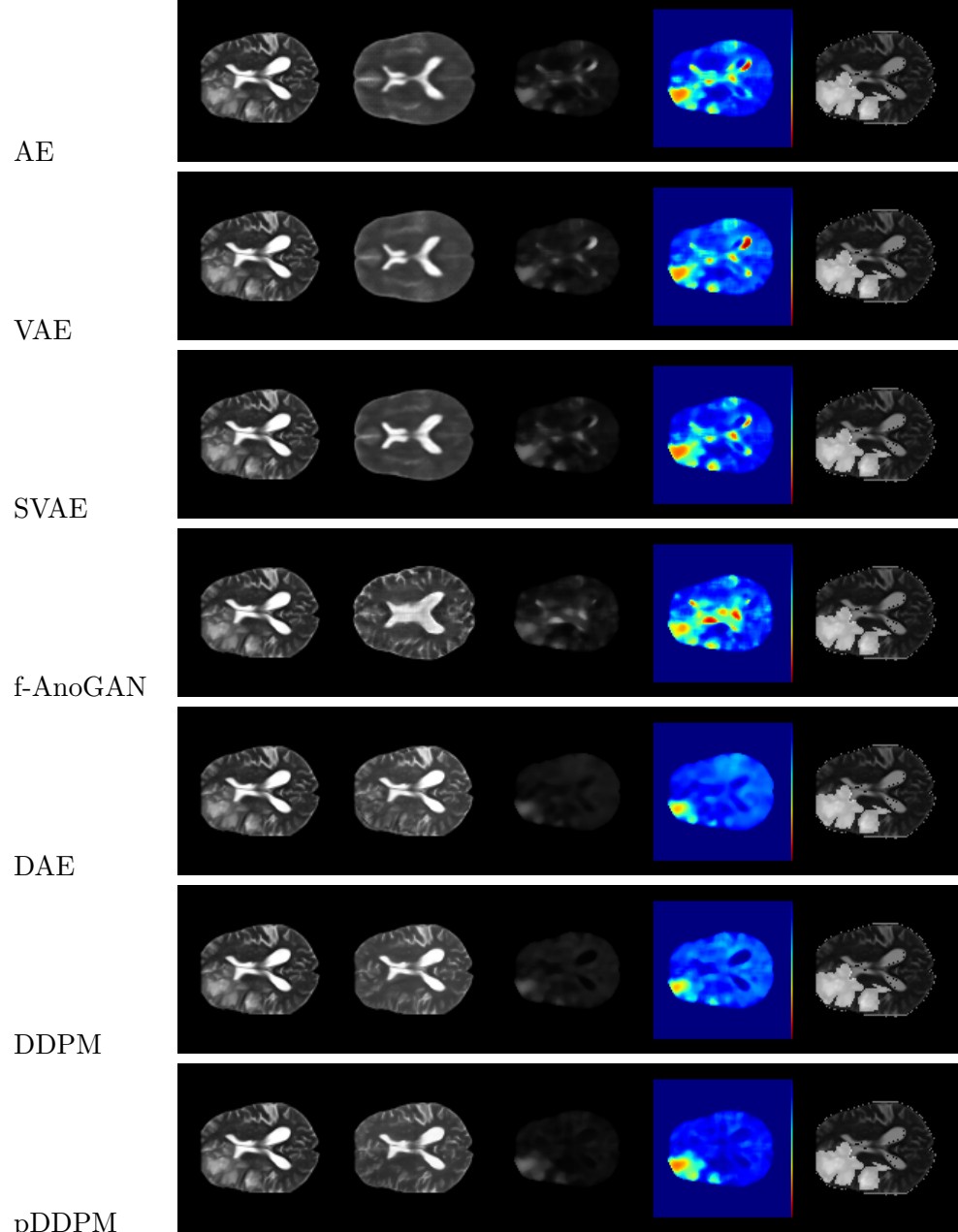

Figure 4: Qualitative evaluation of reconstructions from different models. From top to bottom: AE, VAE, SVAE, f-AnoGAN, DAE, DDPM and pDDPM are presented. From left to right, input, reconstruction, errormap, a heatmap of the errormap and the ground truth annotation is shown

## Appendix B. Visualization of different noise levels

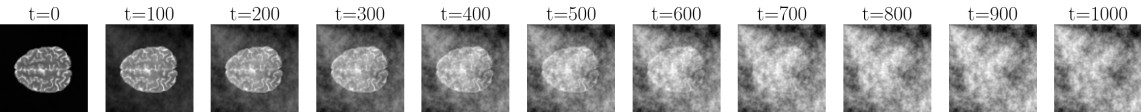

Figure 5:  Training image from the IXI data set pertubed by simplex noise for different time steps $t = 0, 100, ..., 1000$

## Appendix C. Ablation Studies for MSLUB

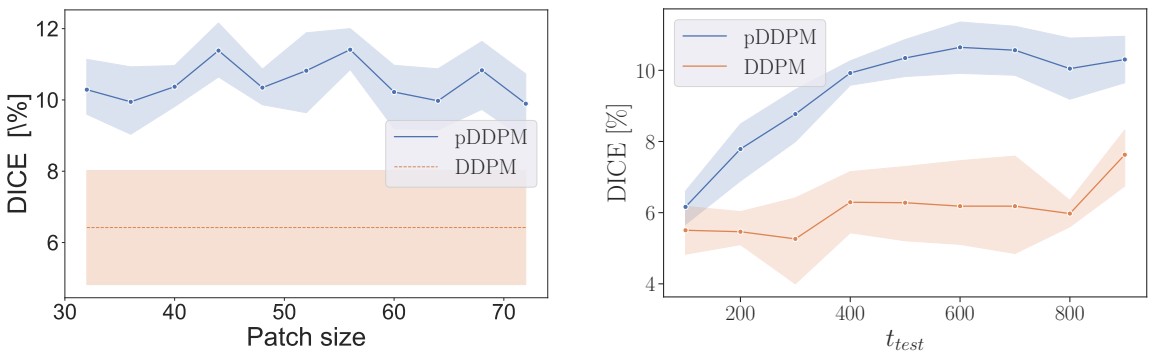

Figure 6:  DICE for different patch sizes (left) and noise levels at test time $t_{test}$ (right) for the MSLUB data set. We report the average DICE across the 5 cross-validation folds. Standard deviations are visualized as enveloping intervals.

