# OpenReview forum: "Patched Diffusion Models for Unsupervised Anomaly Detection in Brain MRI"
_MIDL.io/2023/Conference — MIDL 2023 Poster_

### Official Review · Reviewer_y5Lz · 2023-02-03

**Confidence:** 4
**Preliminary Rating:** 4

**Summary:**

The authors present a method for unsupervised anomaly detection DDPMs. They present a modification on previous reconstruction-based works by only noising patches of the image, whilst feeding the whole image to the DDPM, enabling the denoising process in that patch to be influenced by the full image. At test-time, full-image reconstruction is performed in a sliding window fashion. The authors present results on two datasets, showing some improvements over prior work.


**Strengths:**

- The proposed idea makes sense; allowing reconstruction of a patch with the full image context should lead to better reconstructions of healthy brain
- The method seems to work well
- The paper is clear and easy to follow


**Weaknesses:**

- The methodological innovation is minor: the authors move from the full-image based method of Wyatt et al.
- The method itself bears a lot of similarity to DDPM-based inpainting based methods such as RePaint [1] and I think the authors should acknowledge and discuss this

[1] Repaint: Inpainting using denoising diffusion probabilistic models

**Deanonymize Review:**

no

**Detailed Comments:**

I wonder if the method might further be improved using an iterative refinement process, as done in RePaint [1]

The inference time of their method increases linearly with number of patches, which isn't inconsiderable. Authors should discuss this.

It isn't clear why age stratification was used in the training set.

I would like to see some discussion of what happens when the area outside the patch is unhealthy, or if the edge of a patch intersects a lesion/tumour - does the method reconstruction a lesion within the patch in that case?

The method requires tuning of the T value chosen to denoise from. The results show this optimal value varies depending on the dataset. This tuning could be done on the validation set for each dataset, but another interesting option would be to aggregate the reconstructions from multiple time steps, something like done in [2].

It would be valuable to check and report whether the use of simplex noise is still advantageous in this patch-based scenario when compared to the Gaussian noise used in most DDPMs.

[2] Denoising Diffusion Models for Out-of-Distribution Detection, Graham et al

**Paper Type:**

methodological development

**Questions To Address In The Rebuttal:**

I would like the authors to acknowledge and discuss the overlap of this work and some of the inpainting literature.

I think the authors need to discuss more the need for the choice of a dataset depenedent hyperparemeter, the T value to reconstruct from, as a limitation of the method.

---

### Official Review · Reviewer_8pDs · 2023-02-03

**Confidence:** 4
**Preliminary Rating:** 3

**Summary:**

The paper presents a method of patching DDPMs for improved unsupervised anomaly detection in brain MRI scans. The approach is trained and evaluated on the IXI, BraTS21 and MSLUB datasets. It outperforms standard methods and non-patch-based DDPMs at anomaly detection tasks.

The paper is well-written and tackles an important application. The method is novel and results are promising while there is a weakness in the presentation of the statistical significance tests.

**Strengths:**

- Paper is well organised and written. It is easy to follow and sections are intuitive. The figures are clear with good captions.
- Method is mostly explained clearly (apart from statistical tests, see weaknesses)
- Improvements are notable over comparable methods with repeated experiments.

**Weaknesses:**

- Although the literature review on anomaly detection is good, there is a lack of discussion of patch-based methods. [1], [2] both use patch based diffusion techniques. Although they are slightly different that the method proposed here, it would be proper to acknowledge and discuss the differences. [3] is another patch-based approach for brain MRI analysis, although not using diffusion.
- The authors perform statistical significance tests but do not disclose any details on the method or results, simply state "p<0.05", making it impossible to scrutinise the work. What test was used? How was it used? Was it corrected for multiple testing? What was the outcome?

[1] Özdenizci, O. and Legenstein, R., 2022. Restoring Vision in Adverse Weather Conditions with Patch-Based Denoising Diffusion Models. arXiv preprint arXiv:2207.14626.

[2] Zhang, B., Li, S., Feng, G., Qian, Z. and Zhang, X., 2022, June. Patch Diffusion: a general module for face manipulation detection. In Proceedings of the AAAI Conference on Artificial Intelligence (Vol. 36, No. 3, pp. 3243-3251).

[3] Bintsi, Kyriaki-Margarita, et al. "Patch-based brain age estimation from MR images." Machine Learning in Clinical Neuroimaging and Radiogenomics in Neuro-oncology: Third International Workshop, MLCN 2020, and Second International Workshop, RNO-AI 2020, Held in Conjunction with MICCAI 2020, Lima, Peru, October 4–8, 2020, Proceedings 3. Springer International Publishing, 2020.

Note that two of them are arxiv preprints and had not been peer-reviewed at the time of submission. The MIDL guidelines are not clear as to whether pre-prints should be considered or not so I defer to the AC if this should be considered as part of acceptance criteria.

**Deanonymize Review:**

no

**Detailed Comments:**

- In the introduction: it is written as "denoising probabilistic diffusion models" should be "denoising diffusion probabilistic models"
- The data in Experimental setup is described as 3D volumes but is then sliced to perform patching. What randomisation method was used to selecting slices? Was it with or without replacement?
- Inconsistent notation: sometimes 1e-04 sometimes 10^{-4}
- Add methods for the statistical tests. Was there adjustment for multiple testing?

**Paper Type:**

methodological development

**Questions To Address In The Rebuttal:**

Apart from all the detailed comments above I would like to see additional information and clarification in two key areas:
- How were the statistical tests performed? This should be a separate methods subsection with all details included.
- A deeper discussion on patch-based methods. This paper combines patching with DDPMs and discussing other patching methods and how they are different to this method will be very valuable in order to place this paper in the literature.

---

### Official Review · Reviewer_Pe9E · 2023-02-06

**Confidence:** 4
**Preliminary Rating:** 3
**Recommendation:** Poster

**Summary:**

The authors proposed a novel method for unsupervised anomaly detection (UAD) using 		 	 	 	 denoising diffusion probabilistic models (DDPM) and reconstructing the input images in patch-based manner. The work follows the strategy of detecting anomalies using reconstruction errors as in previous works, and reformulates a model with high image generation capability, patched DDPM (pDDPM), to learn on healthy images and detect anomalous regions with reconstruction errors. The method is evaluated on T2-weighted MRI scans of 2 datasets, and compared to the previous methods.

**Strengths:**

The paper is overall easy to follow and explored the potential of a diffusion model on the task of unsupervised anomaly detection. The authors presented sufficient comparison against existing methods as well as the effects of different hyperparameters for the proposed model. The patch-based reconstruction could also reduce the possible impact of anomalous regions on the healthy regions during reconstruction.

**Weaknesses:**

- Does post-processing using a median filter remove anomalies of small sizes? Such as in Fig.2, the post-processed image appears to be smoother than the original reconstruction error map, if it could be an issue, what should one be aware of tuning the parameter of kernel size for this filter? How generalizable is it if the method is to be applied to MRI scans of other issues/organs?
- Patch size is another hyper-parameter that could affect the accuracy. Is there a way to determine a roughly suitable patch size without acquiring test data with labels or with a little amount of labeled data?
- How does the method work on other modalities, such as T1 and FLAIR? As reconstruction-based error maps are mostly effective for hyper-intense anomalies, e.g. in T2 scans, T1 and FLAIR may contain intensity alteration, will it be more challenging for the detection? Have the authors also considered other losses that can account for structural alteration besides intensity changes?
- Minor issue, Fig.2 seems not to be cited in the paper, on which dataset are the images generated?


**Deanonymize Review:**

no

**Paper Type:**

both

**Questions To Address In The Rebuttal:**

- Please discuss side-effects of the median filter in post-processing of the reconstruction error maps.
- Provide some insights on how to choose suitable hyper-parameters of the proposed method, such as patch-size, noise level, without extensively testing on labelled datasets.
- Please discuss the performance on other modalities of T1 and FLAIR.
- Fix some minor issues and proofread, such as Fig.2 is not cited, and the abbreviation of DDPM should be "denoising diffusion probabilistic models", as in "Recently, denoising probabilistic diffusion models (DDPM) (Ho et al, 2020) have emerged ..." of page 1.

---

### Meta-Review · Area_Chair_Fr5e · 2023-02-22

**Recommendation:** Accept (Poster)
**Confidence:** 5

**Metareview:**

This is a neat application to (patched) DDPMs to anomaly detection in brain MRI. Specifically, it uses a (DDPM) generative model of healthy brains to "project" (potentially) non-healthy tissue onto the model to create a "healthy" version and then estimate anomalies as the voxel-wise difference. The reviewers had minor concerns about clarity, details of the methodology, and statistical analysis of the results, but these were all convincingly addressed in the rebuttal. There were also slightly more serious concerns about the novelty, but these were also well addressed by the authors.

After the rebuttal stage, and despite not being the most novel paper, this is a solid submission with an interesting/relevant application of modern DDPMs that is likely to spark interesting discussion at the conference.